# The Age of the Expert—COVID-19, Expertise, and Conflicts of Interest in Austrian Media Reporting

**Johannes Scherling *** and **Anouschka Foltz**

Department of English Studies, University of Graz, 8010 Graz, Austria; anouschka.foltz@uni-graz.at
* Correspondence: johannes.scherling@uni-graz.at

**Abstract:** Background: Experts are a favorite source of information in the news media as they have the ability to provide balanced and authoritative comments on important issues. However, two factors cast doubt on the extent to which such experts can actually provide balanced information: conflicts of interest and areas of expertise. In this paper, we analyze the use of expert voices during the COVID pandemic in two Austrian broadsheet papers. Methods: We examine the use of reporting verbs employed to indicate the journalists' stance towards the expert comments as well as the relationship of those comments to the experts' fields of expertise and to any potential conflicts of interest. Results: Our analysis shows that the media uncritically reported experts that had considerable conflicts of interest, while others were permitted to comment on topics far outside their particular fields. Conclusions: In the absence of journalistic scrutiny, distance, and context, both of these practices are likely to have led audiences to take the experts' comments at face value and therefore to have embraced unbalanced information that amplified official narratives, to the exclusion of alternative voices.

**Keywords:** news media; experts; COVID-19; Austria; reporting verbs; expertise; conflicts of interest

"Of course all scientists agree, when you censor those who don't."

—Author unknown.

## 1. Introduction

The journalistic rule of objectivity demands that the views of reporters should not be featured in any of their news reports. Reporters, therefore, make frequent use of sources to avoid being perceived as subjective or biased (Sigal 1986) as well as to imbue news reports with an air of credibility. Generally speaking, the media have a particular predilection for what is called official sources—governments, courts, the police, to name just a few. The reason is simple: official sources are deemed credible because of their status, and therefore their information is rarely seen to require vetting (Herman and Chomsky 2002, p. 19). This is also reflected in overly self-confident government statements such as that by New Zealand's then-prime minister Jacinda Ardern during the early days of the COVID crisis in 2020 that "[w]e will continue to be your single source of truth" and that "unless you hear it from us, it is not the truth" (Liles 2022), which was then uncritically reported (see, e.g., Cheng 2020 or, even a year later, Daalder 2021). Officials, aware of the journalists' dependence on their information for making news, use this position of strength "to muster and maintain support [...] for themselves and their preferred courses of action" (Sigal 1986, p. 22). The indexing paradigm (Bennett 1990, p. 106) suggests a further reason for the extensive use of official sources: "Mass media news professionals [...] tend to 'index' the range of voices and viewpoints in both news and editorials according to the range of views expressed in mainstream government debate about a given topic." This means that alternative voices will only surface if they are already established and circulating among elites, which leads to a press that contents itself with the "comfortable role as 'keeper of the official record' while abdicating its traditional mandate to raise an independent 'voice of the people' under appropriate circumstances" (Bennett 1990, p. 106). The resulting media bias

in favor of elites has been well established by decades of media research (e.g., Bagdikian 2004; Davies 2008; Edwards and Cromwell 2006; Entman 2004; Herman and Chomsky 2002; MacLeod 2019; Zollmann 2017). Such biases result in the legitimization of existing power structures (Soloski 1989). "[B]y focusing on elites," Berkowitz and Beach (1993, p. 11) maintain, "the mass media tend to reinforce the dominant ideology. Journalists do not necessarily cause this by conscious intent, though, but largely by the execution of routine journalistic practices".

### 1.1. Experts in the Media and Journalistic Routines

Another favored source of information for interpreting news events are so-called experts, which may feature under labels such as 'senior fellow', 'research director', 'medical expert' or 'political scientist' to provide reports "with a ring of academic respectability" (Davies 2008, p. 171). These are often co-opted by "putting them on the payroll as consultants, funding their research, and organizing think tanks that will hire them directly and help them disseminate their messages", and, as such, they form part of the five filters proposed by the Propaganda Model (Herman and Chomsky 2002, p. 23). In what Henry Kissinger in 1966 called "the age of the expert", these individuals "elaborat[e] and defin[e] [societal] consensus at a high level" and are a product of elite interests.

According to MacLeod (2019, p. 55), "Elite centres of power also create well-funded think tanks to act as official sources that promote policies that further their influence". Such official sources and expert voices, according to a study from 2013, make up the bulk of sources in news media, namely between 60–70% depending on the nature of the media outlet (online/print/TV) (Curran et al. 2013, p. 886), a trend that has been going on since at least the 1960s (Soley 1994, p. 65). Studies by Bauer (1995) and Albaek et al. (2003) further support this claim. In a study on the Syrian conflict, Boyd-Barrett quotes the following from a report by the *Public Accountability Initiative*:

> Twenty-two commentators presented as experts during the so-called corporate media debate about military attacks on Syria have ties to "large defence and intelligence contractors like Raytheon, smaller defence and intelligence contractors like TASC, defence-focused investment firms like SCP Partners, and commercial diplomacy firms like the Cohen Group," the report finds. Of 111 appearances in major media outlets, the ties of these 22 commentators were disclosed a total of 13 times. A majority of these commentators voiced support for a US-led attack on Syria. (Boyd-Barrett 2019, p. 91)

Thus, while journalists may use expert sources to boost their image of objectivity, such experts "often have decidedly partisan or political perspectives that are not acknowledged" (1995, p. 803). Not surprisingly, the financial or professional ties of experts co-determine the content and line of reasoning of their comments (Roussel and Raoult 2020). A problematic aspect involved in this practice is that sources, i.e., the people who are given the right to speak, are afforded the power and privilege to frame the narrative. This is particularly the case if their voices are in the majority and alternative voices are rarely heard, but also when journalists "go beyond simply reporting a view and directly endorse it" (Philo and Berry 2011, p. 176), such as by using a reporting verb like 'state' instead of 'claim' (e.g., 'She stated the situation was dramatic.'), or even no reporting verb at all ('The situation was dramatic'). By repeatedly quoting the same officials and experts, the news media first confer and subsequently reaffirm them as authoritative voices (Cook 2005, p. 92). Selecting experts who mostly echo similar or identical viewpoints then results in the illusion of 'the one truth' held by elite groups through which narrative power is projected. Or, to put it in Tuchman's (1980, p. 250) terms, "power may be realized through the dissemination of some knowledge and the suppression of other ideas".

The choice of experts by individual journalists can be a very complex process and depends on a variety of different factors. Goodell (1977, p. 4) identified "compatibility with the 'media logic'" and "activities outside the scientific community" as two aspects determining the selection of an expert. Leidecker-Sandmann et al. maintain that expertise

as well as "local or regional proximity [...], institutional influence, the scientific actor's availability, and their willingness to express a clear opinion and to provide concise information" (851) are important factors as well. According to Wien (2014), journalists tend to contact researchers that (a) they are already familiar with or (b) that they have seen quoted by other journalists. Lehmkuhl (2021) agrees, saying that "[probably] one of the most important reasons why journalist A selects a scientific expert for news coverage at time X is because journalist B chose the same expert at an earlier time Y." The idea is that if an expert has been vetted by another journalist, then they must be safe to use. The downside may be what Wormer (2020, p. 468) called "a scientist personality cult" at the expense of dissenting voices. Scientific standing, on the other hand, seems to have a limited impact when it comes to choosing an expert. This appears to indicate that the established media presence of experts and their ability to communicate complex information to a general audience may be among the most important factors. Steele (1995, p. 805) claims that journalists' understanding of what constitutes an expert is based not only on experience and access to influential players but also on the expert's willingness to engage in predictions.

It is important to mention, though, that the choice of sources—the choice of experts to be featured in news reporting—is not a choice that journalists are entirely free to make; it is much more dependent on the constraints imposed by news gathering processes and media routines. Due to being allocated to particular topic areas, they will have more frequent contact with some sources and not with others. These will then naturally constitute the go-to group when a story deadline approaches (Sigal 1986, p. 16). Because of these frequent contacts, the relationship between a journalist and their sources will have become "routinized" and "'inextricably intertwined' in a mutually beneficial relationship" (Carlson 2012, p. 4). Such routine sources will then tend to be "favorably portrayed in the news", as journalists have an interest in maintaining their access to these sources (Sigal 1986, p. 28). Furthermore, as Neresini et al. have suggested, "some newspapers may prefer specific experts as media actors according to their ability to align to editorial choices" (2023, p. 18).

### 1.2. Expert Biases and Credentials

Not everyone called an expert necessarily has all the credentials required to comment on the topic in question. In the context of the COVID-19 pandemic, Ioannidis et al. (2021, p. 1) investigated highly visible COVID-19 experts in the media in the US, Denmark, Greece, and Switzerland and concluded that "there is a worrisome disconnect between COVID-19 claimed media expertise and scholarship". For example, only a minority of highly visible COVID-19 experts had published anything on COVID-19. In contrast, Leidecker-Sandmann et al.'s (2022) study suggests that media experts in Germany had above-average expertise in their field. Goswami (2020) maintains that there are certain commenters who are presented as sweeping experts, "consultants and commentators with views on everything under the sun", and will comment on a variety of issues. In fact, as Soley posits, sometimes such experts have very weak credentials (Soley 1994, p. 66). That means that they are given the authority to comment on an event or topic which their experience is not sufficient to warrant commenting on. For instance, just because an expert is a professional doctor does not mean that they are equally able or licensed to comment on all things medical. A virologist, therefore, is not necessarily qualified to comprehensively comment on the intricacies and necessity of lockdown measures. This "move away from a positivist scientific paradigm", according to Boyce, is attributable to "broadening the range of topics on which someone becomes an expert" (Boyce 2007, p. 809).

The practice of primarily anointing individuals as experts that concur with and promote dominant narratives may ignore alternative voices with valuable expertise (Goswami 2020) because their ideas are incongruent with the narrative being advanced and thus not part of the 'indexed' selection of views (Bennett 1990). For instance, in foreign policy stories, people with expert knowledge but with anti-interventionist viewpoints will rarely be heard or labeled an 'expert' or—what is worse—even get attacked and slandered (Klarenberg and Miller 2022; Robinson 2022). This arguably may lead to a sort of expectancy bias by the

media audience that, to qualify as an expert, a person ought to have predefined and not overly incongruous opinions about the agreed-upon positions of governments and elite circles. When alternative experts are employed, they can be merely 'token voices' to suggest balance and are sometimes subject to either name-calling (e.g., Assadist, Putin-apologist, conspiracy theorist, covidiot; see, e.g., Johnstone 2019; Cook 2022; Mark 2020; Brickner 2021) or otherwise labeled with qualifying expressions or modifying words or phrases that insinuate financial or political biases (such as 'Kremlin-backed' or 'close to the XY government'; Rubinstein 2020).

Approved experts, conversely, are more likely to receive legitimacy-conferring modifiers that boost their credibility, such as 'vaccine expert', 'leading researcher', 'internationally renowned', etc. Davies (2008, p. 170) calls such approved expert sources "pseudo-experts" in that they are presented as neutral, objective, and disinterested sources with labels such as 'senior fellow' or 'research director' when in fact they might have a vested interest in promoting the official narrative through financial ties. In his book, *Flat Earth News*, he provides a number of examples of experts with such ties to think tanks and the industry who are presented as neutral 'experts' on the news, when actually they are on the payroll of the very same entities they are meant to comment on (Davies 2008). In the US, for instance, many military analysts on the major networks either have financial ties to the defense industry or are former high-ranking officers in the Pentagon, the military, or in intelligence services like the CIA (Grim et al. 2021). It comes as no surprise, then, that the narratives advocated by such experts are advantageous to the organizations they are financed by or work(ed) for. However, since such experts are neither labeled as providing biased information nor framed as representatives of government or business interests, viewers or readers will more likely consider their opinions to be objective statements of fact (Page et al. 1987). Finally, an issue with such 'authorized' experts is that they create "an objectified image of politics, where citizens are given an interpretation of events rather than being allowed to draw their own conclusions" (Soley 1994, p. 66), such that "editors have come to believe that their highest duty is not to report, but to instruct, not to print news but to save civilization" (Lippmann [1920] 2008, p. 7). This tendency might have intensified following the emergence of television as a more visceral channel to describe the news, leading to a shift towards "interpretive and investigative news journalism" (Albaek 2011, p. 336/7).

### 1.3. Conflicts of Interest in Medical Reporting

The outlined issues are similar and—considering what transpired during the COVID crisis—even more pressing when it comes to news discourse on medical topics. Public organizations such as the CDC (ASH Clinical News 2019) or the FDA (Jewett 2022) receive funding from pharmaceutical companies—something called 'corporate capture' (ESRC-Net n.d.)—where the information coming from them gets compromised to promote pro-pharma talking points at the expense of drug safety and efficacy. Many medical experts—some of whom are featured in mainstream news media—receive funding from or even work for big pharmaceutical corporations such as Merck, Pfizer, or Novartis (Karanges et al. 2020; Prasad 2019). For example, in the first six months of the COVID-19 pandemic, a majority of medical experts most frequently appearing on Japanese television received payments from pharmaceutical companies (Murayama et al. 2021). Others may have been at conferences sponsored by such corporations or may have been otherwise approached by them or received smaller or larger benefits in the form of dinner invitations, complimentary gifts, or even sponsored trips (Fugh-Berman and Ahari 2007; Goldacre 2014; Orlowski and Wateska 1992; Sahm 2013; Steinbrook 2017). This is problematic in that, according to Mintzes et al. (2018, p. 9), "even small gifts, such as food and drink, can affect behaviour", such that, according to US transparency reports (DeJong et al. 2016), "physicians who receive one or more sponsored meals of <US$20 were more likely to prescribe the promoted [drug] product, with larger effects observed the more meals received" (Mintzes et al. 2018, p. 9). Choudhry et al. (2002) have described close relationships between big pharma and

academic researchers and physicians to help promote and market their products. The investigative news site *The Intercept*, for instance, has reported on the case of experts appearing in the media having ties to the pharmaceutical industry without this being revealed, pushing against the sharing of vaccine patents, an issue of great concern to the industry as it might negatively impact their profit margins (Fang 2021). It is not by coincidence when Wien (2014, p. 434) states that "the objectivity of researchers working within, for example, the medical industry is frequently questioned" and they are therefore used only very infrequently (in her case, in Danish media). While claims of experts with ties to industry or official organizations are not necessarily wrong, concerns about such experts' objectivity are widespread, and the evidence that does exist suggests that these concerns are warranted (Bailey et al. 2011; De Dobbelaer et al. 2017; Goldacre 2014; Lexchin 2012; Moynihan et al. 2020; Roussel and Raoult 2020; Sismondo 2021). This raises the question of how many medical experts quoted in the news media have undisclosed ties in some fashion to the pharmaceutical industry and what impact this has on their purported stance as disinterested and objective purveyors of medical and scientific facts. In addition, there has been a certain tendency for journalists and opinion writers to become multipliers and promoters of pharma products (Taschwer 2023; Fox 2021; Freedhoff 2023; Der Standard 2021; Scher 2012), highlighting only the benefits but not the potential harms of the products reported (Moynihan et al. 2000).

In the context of the discourse around COVID-19, therefore, the question beckons to what extent experts quoted in the media have some form of relationship to drug companies and whether this may have affected their stance and interpretation of the 'state of the art' of what was generically referred to as 'the science'. Sismondo suggests that using their considerable resources, "pharmaceutical companies co-opt medical knowledge systems for their particular interests, interests that conflict with the integrity and at least some of the central goals thought to lie behind medicine" (2021, p. 2). For example, "if a pharmaceutical company funds a trial, the chances of results and conclusions in that company's favor are increased", and "funding and payments to researchers create conflicts of interest which [. . .] affect their actions, their judgments, and their conclusions," adding that "[a]s a result, these conflicted researchers become more likely to report outcomes friendly to their funders" (Sismondo 2021, p. 2). What this means for medical experts quoted in the media is obvious: if they have ties to or receive funding from the pharmaceutical companies whose products they are called to comment on, it is more likely than not that their financial involvement with said companies affects the content of their media comments (cf. De Dobbelaer et al. 2017; Goldacre 2014; Mintzes et al. 2018; Roussel and Raoult 2020; Sismondo 2021). This has been observed during the COVID-19 crisis in various contexts. For example, the more money doctors in France received from Gilead Sciences, the manufacturer of the patented COVID-19 treatment Remdesivir, the more they publicly opposed an alternative off-patent COVID-19 treatment (Roussel and Raoult 2020).

It is clear that in a global crisis, as was the case with COVID-19, the public depends on receiving accurate, balanced, and disinterested commentary in the public interest. As Bagdikian states, "[i]ndependent documented information is most needed at the time when officialdom announces a crucial decision. That is when the audience is paying full and anxious attention to conflicting views being debated" (2004, p. 82). In a time of crisis, therefore, being confronted with the full scope of opinions as well as getting information from independent sources without any vested interests in an impending or unfolding crisis is vital to allow the public to support or resist decisions being taken by their governments. Berkowitz and Beach (1993, p. 11) argue that "[f]or conflict stories [. . .] journalists feel compelled by the ideals of objectivity to search out and report on both sides." Similarly, Sigal (1986, p. 16) claims that "[i]n matters of controversy, [journalists] attempt to balance sources with conflicting perspectives." However, Bennett maintains that this 'marketplace of ideas' "tends to disappear at precisely those moments when it would be the most useful for maintaining the democratic balance in the culture" (1990, p. 23). Investigating which of the two claims applies to COVID-19 reporting is therefore one of the goals of this paper.

Lastly, the title of 'expert' is often given not solely or even primarily on the basis of expertise and inside knowledge, but more so with regard to alignment with official positions or official policy goals (see, e.g., Hollar 2022; Herman and Chomsky 2002, p. 24). According to Guy Debord, "All experts serve the state and the media and only in that way do they achieve their status. Every expert follows his master, for all former possibilities for independence have been gradually reduced to nil by present society's mode of organisation" (1988, p. 16/7). This means that whoever gets bestowed the title of 'expert' in the news media is already likely to agree with mainstream narratives, and their place therefore becomes less to provide objective information than to enforce dominant beliefs through the prestige of their title and position. According to Pabst, this expert class "comprises a variety of persons whose societal roles ultimately exist to defend and perpetuate spectacle" (Pabst 2023). In a sense, such experts could therefore be seen as official sources by another name; by quoting them, media outlets signal they are relying on empirical expertise, when in fact they might just be quoting—in disguise—the powers that be. And considering that, as Sigal states, "[n]ews is, after all, not what journalists think, but what their sources say" (1986, p. 29), there is hence a certain probability that official narratives will also be relayed when no official is quoted.

### 1.4. The Current Study

While there has been considerable research performed on the use of sources by journalists (Herman and Chomsky 2002; Albaek et al. 2014; MacLeod 2019), the role of experts in the media (Albaek et al. 2003; Albaek 2011; Wien 2001, 2014), the gender distribution of experts and sources (Prommer and Stüwe 2020; Shine 2021), as well as experts' reactions to audience feedback (Nölleke et al. 2023), there is a paucity of research on potential conflicts of interest or the relationship of experts' professional experience with their specific claims (see also Boyce 2007), in particular for the Austrian context. The little research that does exist on this topic in the context of COVID-19 suggests that experts mentioned in the media tend to have financial conflicts of interest with pharmaceutical companies (Murayama et al. 2021) and that this influences their public statements (Roussel and Raoult 2020). In terms of expertise, previous studies in the context of COVID-19 have yielded conflicting results as to the level of expertise of highly visible media experts (Ioannidis et al. 2021; Leidecker-Sandmann et al. 2022). The main aim of this paper is therefore to contribute to this scarce literature and analyze the use of experts in the Austrian media. We investigate (a) the major experts' potential conflicts of interest, (b) the degree of expertise relative to what was commented on by these major experts, and (c) the linguistic strategies that journalists used to express their stance regarding what these experts have to say. We pose the following research questions:

RQ1: What are the areas of expertise of the experts consulted on COVID-19? Which potential conflicts of interest do the consulted experts have?
RQ2: Which linguistic strategies do journalists use to express their stance towards the experts' statements?
RQ3: How does the experts' expertise relate to the topics that they are commenting on?
RQ4: Which topics do experts with conflicts of interest mainly comment on?

## 2. Materials and Methods

For this paper, we analyzed a corpus of articles from the Austrian newspapers *Der Standard* and *Die Presse*. We selected these newspapers for analysis because of their trusted status and their coverage across a broad political spectrum. Specifically, both are considered to be reputable newspapers in Austria (Kontrast 2018). In 2021, 69% and 67% of people considered *Der Standard* and *Die Presse* to be trustworthy outlets (Gadringer et al. 2022, p. 103). *Der Standard* tends to attract readers from the political left (Gadringer et al. 2022, pp. 64–65), while *Die Presse* aligns itself with conservative and neoliberal perspectives (Eurotopics 2019). Both newspapers appeal to a well-educated readership (MA 2021). *Der Standard* has a readership of approximately 650,000 people (in a country with about

9 million inhabitants), which is higher than *Die Presse*'s roughly 350,000 readers. We focused our analysis on the 10 experts that *Der Standard* and *Die Presse* consulted most frequently in the context of COVID-19.

To collect the articles, we used the metasearch engine WISO and searched for the keywords "Corona", "covid19", and "COVID-19" in the time frame between 1 January 2020, and 31 December 2021 (cf. Scherling and Foltz 2023). We then identified, through a qualitative analysis, the names of reoccurring experts on the topic of COVID-19. These were selected based on the following requirements: (1) They had to be explicitly introduced as 'experts' by journalists, and (2) they were mentioned regularly during the entire time frame we focused on. (3) Their statements had to reasonably inform the news coverage regarding COVID policies, such that readers would be likely to be familiar with them in their roles as experts. After creating this list of the ten most frequently mentioned experts, we did in-depth research on their professional background—their job history, their experience, their funding—in order to detect potential conflicts of interest as well as get an idea of the field of expertise for each of the major experts.

While in a previous study (Scherling and Foltz 2023), we focused on modality and hedging in COVID-19 reporting, for this linguistic analysis, we have looked at strength of verb with regard to the reporting verbs that were used by the journalists. The choice of verb strength in reporting what somebody said is indicative of how a journalist positions themselves toward some claim. There are strong verbs such as "affirm", "establish", and "prove"; more neutral ones such as "discuss", "advocate", and "predict"; and there are weak or doubtful verbs like "propose", "suggest", or "conjecture" (Sowton 2012; Swales and Feak 2012).

Using the corpus software WordSmith 7.0 (Scott 2016), we looked for concordances for the names of the experts, including reporting verbs that were used to qualify their quotes (e.g., "say", "affirm", "claim", "suggest") and the general content of their quotes (e.g., efficacy of masks; necessity to mass-vaccinate). In the case of an interview or whenever an expert was quoted in an article using a colon, we coded that colon as a neutral "say". This was done to see how credible journalists presented these experts' statements and, subsequently, to compare whether the experts' professional experience matched with the expertise necessary to comment on the respective topic or whether there was a case of "weak expertise" (cf. Soley 1994).

Lastly, we also investigated whether the experts' professional background revealed any conflicts of interest with regard to the claims they were making, such as, e.g., promoting a vaccine while being funded by vaccine manufacturers. To gather information about experts' professional backgrounds, we conducted a Google search with their names and consulted their personal/professional websites and other websites that might mention them. We also searched for their names in Google Scholar (using the author option in the advanced search) to ascertain the topics that they had published on, and we consulted the conflicts of interest and funding sections in their academic publications.

## 3. Results

We identified ten major recurring experts in the coverage of COVID-19. Some of these were more frequently quoted than others and gave statements on a greater variety of topics. Mostly, these experts coincided with the members of the government's Corona task force (constituted in early April 2020) or the so-called 'Oberster Sanitätsrat', a body of experts advising the Austrian Ministry of Health. Some were, or at some point, also became members of the COVID task force of experts for the newspaper *Der Standard*.

### 3.1. Expertise and Conflicts of Interest

We begin by looking at the experts' pandemic-related expertise and potential conflicts of interest (research question 1). Table 1 presents the experts most frequently mentioned in *Der Standard* and *Die Presse* in 2020 and 2021, along with their major areas of expertise, basic scientific publishing information derived from Google Scholar, and their potential conflicts

of interest, alongside the number of times they were featured. We have also included an additional expert, Martin Sprenger, who was initially part of the Austrian government's Corona Task Force but then criticized the government's COVID-19 measures and left the task force soon after it was constituted in April 2020 (Anders 2020).

**Table 1.** Commonly mentioned experts along with their major areas of expertise, basic publishing information (i.e., h-index and whether they have published on the COVID-19 pandemic in scientific outlets), their potential conflicts of interest (with reference), and the number of times they were mentioned in the corpus (Standard/Presse).

| Expert | Expertise | Publishing | Potential Conflicts of Interest | Mentions |
|---|---|---|---|---|
| Thomas Czypionka | Health Economics, Health Policy | h-index: 23 COVID: yes | Pfizer (Pock et al. 2008) | 150 (115/35) |
| Christiane Druml | Bioethics, Medical Ethics | h-index: not available COVID: yes | | 99 (79/20) |
| Gerry Foitik | NGO Manager (Red Cross) | h-index: not available COVID: no | | 170 (90/80) |
| Gerald Gartlehner | Clinical Epidemiology | h-index: 69 COVID: yes | World Health Organization (WHO) (Nußbaumer-Streit and Gartlehner 2020), co-convenor of the Cochrane Rapid Reviews Methods Group (Nussbaumer-Streit et al. 2023) | 152 (77/75) |
| Hans-Peter Hutter | Environmental Hygiene, Environmental Medicine | h-index: not available COVID: yes | | 164 (107/57) |
| Peter Klimek | Complex Systems | h-index: 34 COVID: yes | COVID-19 Forecast Consortium, supported by the Austrian Federal Ministry for Social Affairs, Health, Care and Consumer Protection (Bicher et al. 2020) | 222 (76/146) |
| Florian Krammer | Virology, Vaccinology | h-index: 104 COVID: yes | Merck, Pfizer, GlaxoSmithKline, Moderna (Krammer 2020), Seqirus, Avimex, involved in patents/patent applications relating to SARS-CoV-2 serological assays and NDV-based SARS-CoV-2 vaccines (Krammer 2021); research supported by the industry-oriented National Institute of Allergy and Infectious Diseases (NIAID) (Amanat et al. 2020), US Department of Defense (DoD), Bill and Melinda Gates Foundation (BMGF), and National Cancer Institute (NCI); Dynavax (Siegel et al. 2021); Curevac (Carvalho et al. 2021), Third Rock Ventures (Carreño et al. 2022) | 142 (102/40) |
| Nikolas Popper | Computer Modeling, Simulation | h-index: not available COVID: yes | COVID-19 Forecast Consortium, supported by the Austrian Federal Ministry for Social Affairs, Health, Care and Consumer Protection (Bicher et al. 2020) | 280 (179/101) |
| Thomas Szekeres | Genetics, Clinical Chemistry, Laboratory Diagnostics | h-index: not available COVID: yes | Daiichi Sankyo (Bartsch et al. 2022) | 160 (51/109) |
| Dorothee Von Laer | Virology | h-index: not available COVID: yes | ViraTherapeutics GmbH, Boehringer Ingelheim and Pharma KG (Heilmann et al. 2022), Boehringer Ingelheim International GmbH (Moghadasi et al. 2023), inventor on a patent related to VSV-glycoprotein (Das et al. 2021) | 121 (87/34) |
| Martin Sprenger | Public Health, Prevention | h-index: not available COVID: yes | | 80 (73/7) |

The table shows that the expertise of all experts is more or less loosely connected to medicine or public health. Exceptions may be Peter Klimek and Nikolas Popper, who have expertise in complex systems and computer simulation, respectively. Both are areas that do not only apply to medicine and public health. Those experts for whom h-factors were available are prolific researchers in their fields, and all but one expert has published on COVID-19. However, the number of COVID-19 publications varies drastically from expert to expert. Seven of the ten experts, i.e., a majority, have conflicts of interest. We should note that we have included US funding organizations and the WHO as potential conflicts of interest because these organizations have close ties to industry, especially the pharma industry (cf. Ventegodt 2015; Goldacre 2014; Kennedy 2021). We also included Austrian government agencies as potential conflicts of interest because the Austrian government was actively advancing an official narrative. Finally, we considered the Cochrane Rapid Reviews Methods Group to potentially constitute a conflict of interest. While the Cochrane Collaboration strives for independence and transparency, the organization does have ties to the WHO (The Cochrane Collaboration 2022). However, we omitted Austrian and European funding organizations from our list of conflicts of interest as these funding organizations' relationship to industry is less clear. It is further possible that the remaining experts also have conflicts of interest that we did not discover. Overall, all experts have relevant expertise in some fashion, but more than half of them have conflicts of interest that could have led to bias or that may have prevented them from voicing dissenting opinions.

### 3.2. Reporting Verbs

In order to ascertain the degree of trust and credibility awarded to the major experts used in Austrian newspapers for the coverage regarding COVID-19, we looked for the reporting verbs that the media used to introduce any claims brought forward by them (research question 2). If the media used many weak and tentative reporting verbs, that would indicate a critical distance; if, on the other hand, there was a preponderance for more neutral or even assertive reporting verbs, this would suggest that journalists embraced the experts' claims.

The data in Table 2 reveal that the COVID experts' claims were all but embraced as self-evident facts by both *Die Presse* and *Der Standard* journalists. As the table shows, for all experts, unequivocally the most used reporting verb is *sagen* (*to say*), which suggests that reporters are overwhelmingly taking experts' comments at face value or, at the very least, are not questioning them. Other common reporting verbs are *erklären* (*to explain*), *warnen* (*to warn*), and *halten für* (to consider), as well as other lexemes like *so/laut* (both of which approximately mean *according to*). Occasionally, there is no reporting verb, and instead the journalists employ the subjunctive form (such as *seien* for *to be*), which can be used to indicate indirect speech.

There are barely any weak reporting verbs like *vermuten* (*assume*, two times) or *behaupten* (*claim*, zero times), with the sole exceptions of *meinen* (*think*, 38 times) and *glauben* (*believe*, 22 times), which mainly feature when the experts make predictions on possible future developments.

This overwhelming use of neutral and sometimes assertive reporting verbs suggests that the journalists portray the words of the experts as factual knowledge. There is no relevant use of hedging or distancing from their claims in any of the articles that we looked at. Modality only features when the experts make conjectures regarding future developments or occurrences. We also found no counter-claims by experts included to mitigate the opinion monopoly. There is thus a near 100% trust that the information relayed by the chosen experts is empirically founded and based on each expert's expertise on the matter.

What is particularly interesting is the most frequent reporting verb *to say*. For all experts, this is overwhelmingly the most frequent reporting verb, with the exception of Martin Sprenger, who was on Austria's Corona Task Force until he left the task force in dissent. As a result of his critical stance, the verb *to criticize* is the second most used

reporting verb for Sprenger, a verb that is absent from the top three for all other experts. Thus, there is little evidence that the remaining experts took any kind of critical stance, except when calling for even tougher measures.

**Table 2.** The three most-used reporting verbs for each expert.

| Expert | Reporting Verb (N) | | |
|---|---|---|---|
| | **Most Frequent** | **2nd Most Frequent** | **3rd Most Frequent** |
| Czypionka | sagen (93) (to say) | erklären (10) (to explain) | halten für (7) (to consider) |
| Druml | sagen (73) (to say) | seien (12) (indirect speech) | meinen (6) (to think) |
| Foitik | sagen (43) (to say) | seien (8) (indirect speech) | vorschlagen (7) (to propose) |
| Gartlehner | sagen (66) (to say) | halten für (11) (to consider) | so (10) (according to) |
| Hutter | sagen (43) (to say) | so (11) (according to) | seien (7) (indirect speech) |
| Klimek | sagen (88) (to say) | erklären (15) (to explain) | seien (13) (indirect speech) |
| Krammer | sagen (57) (to say) | glauben (5) (to believe) | beantworten (5) (to answer) |
| Popper | sagen (71) (to say) | so (15) (according to) | laut (15) (according to) |
| Szekeres | sagen (28) (to say) | so (9) (according to) | sich aussprechen für (7) (to argue for) |
| Von Laer | sagen (22) (to say) | warnen (8) (to warn) | laut (5) (according to) |
| Sprenger | sagen (7) (to say) | kritisieren (6) (to criticize) | für/so (4) (for/according to) |

*3.3. Weak Credentials*

We now consider how the experts' expertise relates to the topics that they are commenting on (research question 3). Overall, not everything the experts comment on is located squarely inside their special field of expertise. While the medical experts mostly discuss issues that are at least related to the (admittedly large and diverse) field of medicine, there are others, most notably Gerry Foitik, Peter Klimek, Christiane Druml, and Niki Popper, who—in Goswami's (2020) words—at times seem to "have views on everything under the sun", or at least on things beyond their immediate professional experience. We therefore consider these four experts' comments in more detail in this section.

Peter Klimek has a background in theoretical physics and is a researcher in complexity science, aiming "to improve our understanding and ability to predict complex socio-economic systems, ranging from human disease over healthcare systems to economic and financial systems" (MedUni Wien n.d.). As such, he works within the Medical University of Vienna and in a medical context, but is himself neither a medical professional nor a social scientist. Table 1 additionally suggests that he has ties to the Austrian government and might be prone to repeating official government narratives. The nature of some of his comments in the news media, however, is on subjects that can be suggested and are not based on his immediate professional expertise. Notably, he makes various recommendations that have immense social implications for the population. Since he is not a social scientist, such recommendations are arguably outside of his expertise. For example, he recommends curfews, restrictions for large events, and the enforcement of

home office (*Der Standard*, 18 November 2021). He proposes tightening quarantine rules and requiring up-to-date vaccination or recent recovery plus a PCR test that is not older than 48 h for the entire leisure sector nationwide (*Die Presse*, 14 November 2021). All of these recommendations have far-reaching (and mostly negative) implications for people's social lives and mental well-being (Dale et al. 2021; Panchal et al. 2023; Pieh et al. 2021; Serrano-Alarcón et al. 2022). Similarly, he comments on medical topics (often with ethical implications) that are outside of his immediate area of expertise. For example, he discusses whether to have mandatory vaccinations in general or only for certain occupational and risk groups (*Die Presse*, 24 November 2021), supports additional restrictions for the unvaccinated (*Der Standard*, 7 September 2021), suggests that more first vaccinations are needed to avoid a fifth wave (*Die Presse*, 14 November 2021), promotes PCR tests over antigen tests (*Der Standard*, 29 July 2021), and makes claims as to the extent to which the unvaccinated contribute to the incidence of infection (*Die Presse*, 14 November 2021).

Overall, Klimek promotes claims that are not in his personal field of expertise. There is little question that they reflect the published mainstream consensus at that time, but Klimek is likely not the go-to expert on these particular issues. What we have here is less first-hand information based on research but rather second- or third-hand information based on a subjective evaluation of credibility.

A similar situation holds in the case of Niki Popper, who is a simulation scientist and studied mathematics. His research focus is on "theory and applications of modelling & simulation of dynamic and complex systems" (TU Wien n.d.). It could therefore be expected that he comments on what his simulations have shown or makes tentative projections on how certain measures might impact the development of COVID-19 infections. Some of his statements definitely focus on his field of expertise; however, there are also a considerable number of comments that show him going beyond his immediate field of expertise. Similar to Klimek, he also makes statements that have profound (and potentially negative) social implications for people as well as statements within the medical realm, both of which are outside of his immediate area of expertise. For example, he promotes more effective digital contact tracing and quarantine (*Die Presse*, 15 September 2020; *Der Standard*, 21 November 2020). He also discusses the effectiveness of herd immunity (*Die Presse*, 19 February 2021) and the effects of vaccinations on hospitalization rates and disease spread (*Der Standard*, 16 January 2021), as in example (1):

1. „Die Hospitalisierungsrate stieg um 81 Personen auf 1370, 285 davon waren am Freitag in Intensivbetreuung—ein Plus von 15 Personen. Simulationsforscher Niki Popper sieht unter anderem den Rückgang der Impfwirksamkeit als Grund für diesen sprunghaften Anstieg." ("The hospitalization rate increased by 81 people to 1370, 285 of whom were in intensive care on Friday—an increase of 15 people. Simulation researcher Niki Popper sees the decline in vaccine effectiveness, among other factors, as the reason for this jump."; *Der Standard*, 30 October 2021)

Popper is also a strong proponent of vaccinations and makes various strong claims and forceful proposals to increase vaccine uptake, as in examples (2) through (4):

2. „'Die Hauptnachricht muss sein: Impfen, weil das ist das Einzige, das hilft.' Aber das erfordere auch ‚klare Ziele', die die Politik benennen und dann auch verfolgen müsse, betont Popper, der auch dem Covid-Prognose-Konsortium angehört." ("'The main message has to be: Vaccinate, because that's the only thing that helps.' But that also requires 'clear goals' that policymakers must name and then pursue, stresses Popper, who is also a member of the Covid Forecast consortium."; *Der Standard*, 31 August 2021)

3. „'Die Menschen sind nur verunsichert', glaubt Popper: ‚Mit transparenter Kommunikation und einem niederschwelligen Angebot, das Vertrauen schafft, gibt es sicher noch viel Potenzial. Bei jedem Feuerwehrfest muss der Impfbus vorfahren—am besten mit 50 Liter Bier im Gepäck.'" ("'People are just insecure,' believes Popper: 'With transparent communication and a low-threshold offer that creates trust, there is certainly still a lot of potential. The vaccination bus must drive up at every fire

department festival—preferably with 50 liters of beer in its luggage.'"; *Der Standard*, 7 September 2021)

4.   „Popper sagt im STANDARD-Telefonat angesichts dieser Entwicklungen: ‚Impfen muss die oberste Priorität haben.'" ("Popper says in STANDARD phone call in light of these developments, 'Vaccination must be the top priority.'"; *Der Standard*, 26 August 2021)

Again, and as with Klimek, Popper comments on a variety of issues along the lines promoted in mainstream circles, but considering his CV, he does not appear to have the professional expertise necessary to make these statements under the expert label. The question of vaccines, the necessity of quarantine, the question of why infections suddenly rise—all those would need to be answered by the likes of virologists or infectiologists, ideally together with social scientists and ethicists. Coming from Popper, this could only be framed as the personal opinion of an individual, but it is framed like an expert statement.

Another frequently quoted expert, Gerry Foitik, studied business administration and has been active in the Red Cross organization since 2000. He is currently the 'Bundesrettungskommandant' (the federal commander of rescue services) of the selfsame rescue organization, and as such, he holds extensive expertise in the field of operational management (Österreichisches Rotes Kreuz 2023). Nonetheless, a considerable number of his statements concern topics that are rather removed from his professional expertise. Most notably, he makes numerous proposals that are in the medical realm. For instance, he proposes to make lockdowns contingent on specific infection incidence rates (*Der Standard*, 15 December 2021), suggests that vaccinations in Austria had begun too late and too hesitantly (*Die Presse*, 27 April 2021), makes predictions about the future number of COVID-19 infections (*Der Standard*, 22 September 2020), and makes recommendations regarding the testing of people at high risk of being infected with COVID-19 (*Die Presse*, 20 October 2020). Example (5) is particularly interesting in this regard. Here, Foitik suggests that people who are in quarantine for having had contact with an infected individual should not be tested because they need to sit out their quarantine 'until the bitter end' anyway, and too many positive test results might endanger the winter tourism season:

5.   „Ob nämlich Menschen, die Kontakt mit COVID-19-Infizierten hatten und deshalb zehn Tage in Quarantäne müssen, auch dann getestet werden sollen, wenn sie keine Symptome haben. Nein, meint Foitik, sein Argument: Das Testergebnis ändere ohnehin nichts, der Quarantänling muss seine Zeit so und so (positiv oder negativ) bis zum bitteren Ende zu Hause absitzen. Positive Ergebnisse erhöhen aber die Fallzahlen und schaden damit dem Tourismusstandort, weil auf ihrer Grundlage Reisewarnungen gegen Österreich erlassen werden, die gerade die Wintersaison gefährden." ("The question is whether people who have had contact with COVID-19 infected persons and therefore have to be quarantined for ten days should be tested even if they have no symptoms. No, says Foitik, his argument: The test result changes nothing anyway, the quarantined person must serve their time until the bitter end at home regardless of whether they're positive or negative. Positive results, however, increase the number of cases and thus harm tourism, because travel warnings against Austria are issued on their basis, which endanger the winter season in particular."; *Die Presse*, 20 October 2020)

Interestingly, even though in 2020 he promoted keeping people in quarantine regardless of whether or not they would actually test positive for COVID-19, in 2021 he is worried about too many people not being able to come in to work due to COVID-19, as example (6) shows:

6.   „Drastisch wurde am Mittwoch deshalb auch Gerry Foitik, Bundesrettungskommandant des Roten Kreuzes. Er forderte weitreichende Maßnahmen, um die Ausbreitung des Virus einzudämmen. Denn laut Foitik seien die Herausforderungen, vor die das Land durch Omikron gestellt werde, selbst im Best-Case-Szenario ‚gewaltig', schrieb er auf Twitter—wenn zu viele Menschen beispielsweise gleichzeitig nicht arbeiten

können, weil sie infiziert zu Hause bleiben müssen. Foitik zufolge könnte das 30 Prozent aller Arbeitnehmerinnen und Arbeitnehmer betreffen." ("Therefore, Gerry Foitik, federal rescue commander of the Red Cross, also became drastic on Wednesday. He called for far-reaching measures to contain the spread of the virus. That's because, Foitik said, the challenges the country faces from Omicron are 'daunting' even in a best-case scenario, he wrote on Twitter—if too many people are unable to work at the same time, for example, because they have to stay home infected. According to Foitik, that could affect 30 percent of all workers."; *Der Standard*, 15 December 2021)

Similar to the previous two experts, he also discusses topics with profound social implications. In particular, he promotes indoor mask mandates, including in schools and for all events (*Die Presse*, 8 August 2020; *Der Standard*, 15 December 2021). From his professional background, it is not quite clear what the media thought qualifies Foitik to make statements of that sort aside from providing his personal opinion.

Finally, Christiane Druml is another frequently cited expert who, as we will show, gives opinions on topics that do not immediately pertain to her special field of expertise. Druml studied law and later became involved in bioethics, which eventually would result in her appointment to the Austrian National Bioethics Commission (Bundeskanzleramt n.d.) and, later, the COVID task force installed by the Austrian government (Fabry 2020). Her specialty is, therefore, ethical questions relating to medical decisions and research. Most notably, despite her expertise in bioethics, some of her statements that involve ethical considerations seem contradictory. Example (7) illustrates this. Here, she considers an indirect vaccine mandate by means of additional restrictions for the unvaccinated to be an 'unethical argument' (after having supported additional restrictions for unvaccinated people in certain professions about nine months earlier; *Der Standard*, 19 December 2020), but then proposes that such an indirect vaccine mandate is fine from an epidemiological viewpoint. She also suggests that measures should not be implemented with the intention of making life difficult for the unvaccinated (which is exactly what the proposed indirect vaccine mandate does), but then immediately proposes increased pressure on the unvaccinated.

7.  „Für Christiane Druml, Vorsitzende der Bioethikkommission, ist diese indirekte Impf-pflicht ein ‚unethisches Argument'. Es gebe allerdings aus epidemiologischer Sicht genügend Gründe für Einschränkungen von Ungeimpften, die letztlich auf das Gleiche hinauslaufen würden. Allerdings müsse jeder Schritt wissenschaftlich begründet werden und nicht mit der Absicht, Ungeimpften das Leben möglichst schwer zu machen. Und auch Druml hält einen verstärkten Druck auf Ungeimpfte für notwendig. ‚3,5 Millionen ungeimpfte Österreicher sind eine dramatische Situation', sagt sie. ‚Wir können als Gesellschaft so nicht weitermachen. Wir müssen schauen, dass Menschen ihre gesellschaftlichen Pflichten erfüllen.'" ("For Christiane Druml, chairwoman of the Bioethics Commission, this indirect vaccination requirement is an 'unethical argument'. However, from an epidemiological point of view, there are sufficient reasons for restrictions on the unvaccinated, which would ultimately amount to the same thing, she says. However, she said, each step must be justified scientifically and not with the intention of making life as difficult as possible for the unvaccinated. And Druml also believes increased pressure on the unvaccinated is necessary. '3.5 million unvaccinated Austrians is a dramatic situation,' she says. 'We cannot continue like this as a society. We have to see that people fulfill their social obligations.'"; *Der Standard*, 11 September 2021)

The above example also suggests that her ethical perspective is influenced by her medical beliefs. Specifically, she focuses on the rights of the collective rather than the individual when she says that people need to fulfill their social obligations or when she promotes masks, social distancing, and increased hygiene measures because 'On the subway, you don't know if the person across from you is going to a Corona demonstration or has already been vaccinated for the second time' (*Der Standard*, 23 February 2021). This focus on individuals' obligations towards societies rather than on individuals' rights

to bodily integrity or individual choice is likely related to her beliefs about vaccines. Specifically, she considers vaccinations to be "one of the greatest achievements of medicine, which have led to an increase in life expectancy, prosperity and social justice" and are "among the measures that have saved the most lives of all medical interventions" (Druml 2023, p. 121). Ethical concerns for individuals only arise if one considers potential vaccine side effects or adverse events.

Example (8) shows another contradiction, where in the same sentence, she acknowledges that omicron is an unknown but then states that there is no alternative to another lockdown, a measure with both ethical and social implications.

8. „STANDARD: Also ist die Lage so katastrophal geworden, dass es nun nicht mehr anders geht?" „Druml: Ich glaube, sie ist katastrophal geworden. Wenn die Situation so weitergeht—wobei Omikron eine Unbekannte ist -, wird an einem Lockdown am Anfang des Jahres kein Weg vorbeiführen." ("STANDARD: So the situation has become so catastrophic that there is no other way now?" "Druml: I think it has become catastrophic. If the situation continues like this—Omikron being an unknown—there will be no way around a lockdown at the beginning of the year."; *Der Standard*, 20 December 2021)

We can see that she mixes ethical judgments with medical claims in many of her comments. She essentially relies on second-hand medical knowledge or opinions formed through media exposure to promote measures such as discrimination of unvaccinated individuals or harsher measures to contain the pandemic. As with the other experts presented above, the problem is that Druml provides statements and comments that are more comprehensive than her professional expertise extends. This overstretch by her and the other non-medical experts in the two news media we analyzed is never addressed in the articles in our corpus, and, therefore, readers are unlikely to question the legitimacy or credibility of their respective statements. This is enhanced by the uncritical use of neutral and assertive reporting verbs that show no scrutiny regarding the claims of said experts.

### 3.4. Conflicts of Interest

We now turn to the question of which topics experts with conflicts of interest mainly comment on (research question 4). We have identified a number of experts with conflicts of interest above, most notably Florian Krammer. Other experts with connections to pharmaceutical companies are Thomas Czypionka, Thomas Szekeres, and Dorothee Von Laer. We begin with Krammer, who has by far the most conflicts of interest. His comments center around COVID-19 vaccines, and as a virologist and vaccinologist, he remains squarely within his area of expertise. There is, however, evidence that his conflicts of interest may have influenced his views on the vaccine, having himself been involved in vaccine research and funded by some of the companies behind the COVID vaccines. His main function in the media seems to have been to assure readers that the vaccines are 'safe and effective'—an important message for the pharmaceutical industry given how quickly the vaccines were developed. One of his strategies is to talk about side-effects observed for other vaccines and then reassure the public that the COVID-19 vaccines do not have these particular problems (*Der Standard*, 3 February 2021; *Der Standard*, 27 November 2020). When talking directly about the COVID vaccines, his main message is also that of reassurance in their safety, as shown in examples (9) and (10):

9. „‚Wenn man 15 Jahre Zeit gehabt hätte, hätte man vielleicht einen Impfstoff gefunden, der wirksam ist, aber kaum Reaktionen hervorruft. Aber diese Zeit hatten wir jetzt nicht. Die Impfstoffe sind sicher und schützen, aber sie verursachen Impfreaktionen, die zwar ungefährlich, aber teilweise unangenehm sind', sagt Krammer." ("'If we had had 15 years, we might have found a vaccine that was effective but hardly caused any reactions. But we haven't had that time now. The vaccines are safe and protect, but they cause vaccine reactions that are harmless but sometimes unpleasant,' Krammer says."; *Der Standard*, 24 March 2021)

10. Frage: „Bei welchen Allergikern kam es zu heftigen Impfreaktionen, und sollen diese daher nicht geimpft werden?" Krammer: „Die einzigen Allergiker, die aus der Pfizer-Phase-III-Studie ausgenommen wurden, waren Personen, die mit starken allergischen Reaktionen auf Impfungen reagieren. Man kann also annehmen (basierend auf dem hohen Anteil von Allergikern in der Bevölkerung), dass viele Allergiker an der Studie teilnahmen, und es gab keine [sic.] Problem." (Question: "In which allergic persons did severe vaccination reactions occur, and should they therefore not be vaccinated?" Krammer: "The only allergy sufferers excluded from the Pfizer Phase III study were individuals with severe allergic reactions to vaccination. So it's reasonable to assume (based on the high proportion of allergy sufferers in the population) that many allergy sufferers participated in the study, and there was no problem."; *Der Standard*, 14 December 2020)

Example (11) is particularly interesting. Here, Krammer is confronted with a question about a specific RNA vaccine where various side effects, including autoimmune diseases, occurred in animals. The implication of this question is that, if previous RNA vaccines had side effects, could this be an issue with the COVID mRNA vaccines, too? Instead of answering the question, though, he becomes evasive and talks about a different mRNA vaccine and states that autoimmune diseases were not a problem for this particular mRNA vaccine.

11. Frage: „An RNA Impfstoffen wird seit mehr als zwei Jahrzehnten geforscht. Zugelassen wurde noch keiner (abgesehen jetzt von England), u.a. weil es bei Tierversuchen zu verschiedenen Nebenwirkungen wie Autoimmunerkrankungen kam. Wie lange hat es gedauert, bis diese Tiere Autoimmunerkrankungen entwickelt haben?" [. . .] Krammer: „Die ersten RNA-Impfstoffe wurden 2013 an Menschen getestet, und wie das eben normalerweise so ist, mahlen die Mühlen hier langsam. Einige dieser Impfstoffe sind erfolgreich und werden weiterentwickelt. Moderna hat zum Beispiel gerade eine Phase-II-Studie mit einem Cytomegalievirus-Impfstoff gestartet. Im Tiermodell funktionieren diese Impfstoffe hervorragend. So weit ich weiß, waren Autoimmunerkrankungen kein Problem im Tiermodell, aber man muss das natürlich monitoren, vor allem bei Menschen, die zu Autoimmunreaktionen neigen." (Question: "Research on RNA vaccines has been going on for more than two decades. None have yet been approved (except now in England), in part because various side effects such as autoimmune diseases have occurred in animal studies. How long did it take for these animals to develop autoimmune diseases?" [...] Krammer: "The first RNA vaccines were tested in humans in 2013, and as is usually the case, the mills grind slowly here. Some of these vaccines have been successful and are being developed further. For example, Moderna has just started a Phase II trial with a cytomegalovirus vaccine. In animal models, these vaccines work extremely well. As far as I know, autoimmune disease hasn't been a problem in animal models, but you obviously have to monitor that, especially in people who are prone to autoimmune reactions."; *Der Standard*, 14 December 2020)

Even as late as May 2022, after over a million adverse events following COVID-19 vaccination had been reported to the European database of suspected adverse drug reaction reports (EudraVigilance n.d.), Krammer spread the message that the vaccines were safe, as example (12) shows:

12. „Für diese Impfung gibt es sogar mehr Daten als bei vielen anderen, was die Sicherheit betrifft, insofern sind das auf jeden Fall einige der sichersten Vakzine, die wir haben." ("There's even more data for this vaccine than there is for many others in terms of safety, so in that sense, these are definitely some of the safest vaccines we have."; *Der Standard*, 15 May 2022)

In addition to focusing on safety, Krammer also overestimated the duration of efficacy of the vaccines when they were first available, as example (13) from December 2020 illustrates:

13. „Zwar gibt es laut Krammer noch keine gesicherten Fakten. ,Aber aufgrund der Antikörperantwort, die man jetzt sieht, und der Daten, die es bis jetzt gibt, also über vier Monate, nehme ich mal an, dass der Schutz durch die Impfung schon für einige Jahre hält.', sagte er zu orf.at.“ ("Although, according to Krammer, there are no definite facts yet. 'But based on the antibody response that you see now and the data that's out there so far, so over four months, I'm going to assume that the protection from the vaccination will last for several years,' he told orf.at."; *Die Presse*, 30 December 2020)

Krammer's belief in the safety and efficacy of the vaccines seems to also influence his views on related topics, such as whether there should be mandatory vaccination or whether to vaccinate children, as example (14) shows:

14. STANDARD: „In Österreich ist derzeit ein Gesetz für eine allgemeine Impfpflicht gegen Corona in Vorbereitung. Kann das, auch angesichts der Unwägbarkeiten wegen Omikron, die Pandemie unter Kontrolle bringen?“ Krammer: „Ja—obwohl ich grundsätzlich nicht dafür bin, dass man Leute zu irgend etwas zwingt. Ich finde es traurig, dass man eine Impfpflicht einführen muss, aber es gibt anscheinend keine andere Lösung. Ich glaube schon, dass die Impfpflicht sehr dazu beitragen wird, die Situation zu beruhigen. Hätte man im Sommer, also vor der Herbstwelle, mehr Leute dazu motivieren können, sich impfen zu lassen, wäre das Problem mit Corona jetzt viel kleiner. Natürlich, wir konnten da noch keine Kinder impfen—jetzt kann man das und das ist gut, denn viele Kinder infizieren sich und geben die Infektionen weiter. Ich glaube, mit der verpflichtenden Impfung und einer höheren Durchimpfungsrate bei Kindern sind die Aussichten nicht allzu schlecht.“ (STANDARD: "In Austria, a law is currently being prepared for general compulsory vaccination against Corona. Can that, also in view of the uncertainties because of Omicron, bring the pandemic under control?" Krammer: "Yes—although I am generally not in favor of forcing people to do something. I think it's sad that you have to make vaccination compulsory, but there doesn't seem to be any other solution. I do believe that compulsory vaccination will help a lot to calm down the situation. If we had been able to motivate more people to get vaccinated in the summer, before the fall wave, the problem with Corona would be much smaller now. Of course, we couldn't vaccinate children then—now we can, and that's good, because many children get infected and pass on the infections. I think with mandatory vaccination and higher coverage among children, the prospects aren't too bad."; *Der Standard*, 5 December 2021)

Thomas Czypionka's comments about vaccination tend to have a slightly different angle. His main concern is increasing the vaccination rate for COVID and other diseases (*Der Standard*, 29 October 2021; *Die Presse*, 12 December 2020; *Die Presse*, 25 November 2021) and how the government can convince more people to get vaccinated, for example, through mandatory consultation appointments in a vaccination center (*Der Standard*, 13 December 2021) or through vaccination incentives (*Die Presse*, 25 November 2021). However, despite conflicts of interest, he is one of the experts who has acknowledged that adverse vaccine events are underreported, as illustrated by example (15):

15. „Die Kenntnisse über die Sicherheit von Arzneimitteln sind zum Zeitpunkt ihrer ersten Zulassung nie vollständig. Deshalb gibt es in Europa ein System, um auftretende Nebenwirkungen und Impfreaktionen rasch zu melden und bei Bedarf abzuklären. Diese ,Pharmakovigilanz' funktioniert allerdings in der Praxis in den verschiedenen Ländern unterschiedlich, weshalb Gesundheitsexperten wie Thomas Czypionka vom Institut für Höhere Studien davon sprechen, dass es zu Untererfassungen kommt. [. . .] Mit Ausnahme von wenigen Ländern gibt es aber keine systematische Erfassung von Daten. Beispiel Österreich: Hier sieht das Arzneimittelgesetz vor, dass vermutete Nebenwirkungen von Ärzten an das Bundesamt für Sicherheit im Gesundheitswesen (BASG) gemeldet werden müssen. In der Praxis tun das Ärzte im niedergelasse-nen Bereich kaum und auch in den Spitälern nicht systematisch. ,Wir haben keinen Überblick in Österreich, wie sollten wir den auch haben¿, sagt Czypionka.“ ("Knowl-

edge about the safety of pharmaceuticals is never complete at the time of their initial approval. That's why Europe has a system in place to quickly report any side effects or vaccine reactions that occur and to check them if necessary. However, this 'pharmacovigilance' works differently in practice in different countries, which is why health experts such as Thomas Czypionka of the Institute for Advanced Studies say that under-reporting occurs. [...] With the exception of a few countries, however, there is no systematic collection of data. Take Austria, for example: Here, the Medicines Act stipulates that suspected adverse reactions must be reported by physicians to the Federal Office for Safety in Health Care (BASG). In practice, physicians in private practice hardly do this, nor do they do it systematically in hospitals. 'We don't have an overview in Austria, how could we?' says Czypionka."; *Der Standard*, 18 March 2021)

Thomas Szekeres is one of the strongest proponents of the government's general vaccine mandate for everyone 14 years and up (*Die Presse*, 15 November 2021; *Die Presse*, 4 December 2021), though he would consider it sensible to vaccinate everyone five years and up (*Die Presse*, 4 December 2021). He also supported vaccinating children off-label before the vaccine was officially recommended for that age group, as example (16) shows:

16.    „In Europa wird noch auf die Zulassung dieses Impfstoffs durch die Arzneimittelagentur EMA gewartet. Ärztekammer-Präsident Thomas Szekeres rechnete mit dem offiziellen Go für die Kinderimpfung bereits "in den nächsten Tagen". Das Angebot der Stadt Wien ‚off label' befürwortete Szekeres." ("In Europe, the approval of this vaccine by the EMA drug agency is still pending. Medical Association President Thomas Szekeres expected the official go for the children's vaccination 'in the next few days' already. Szekeres endorsed the City of Vienna's 'off label' offer."; *Der Standard*, 15 November 2021)

He also supports numerous measures that discriminate against unvaccinated people, such as lockdowns for the unvaccinated (*Die Presse*, 15 November 2021), sending unvaccinated employees home without pay (*Die Presse*, 14 November 2021), and compulsory COVID-19 testing at one's own expense for unvaccinated individuals who want to attend events or go to a restaurant (*Die Presse*, 5 August 2021). Example (17) illustrates that the 'safe and effective' narrative is his reason for promoting compulsory COVID-19 testing at one's own expense for the unvaccinated and possibly other discriminatory measures:

17.    „‚Ich sehe nicht ein, warum die Allgemeinheit diese doch sehr kostspieligen Tests zahlen muss, wenn Menschen sich weigern, die Gratis-Impfung in Anspruch zu nehmen. Die Impfungen sind von der europäischen Zulassungsbehörde streng kontrolliert worden, sind bei uns zugelassen, sind sicher und wirkungsvoll—und ich glaube, es sollte sich jeder nach Möglichkeit impfen lassen.'" (" 'I don't see why the general public has to pay for these tests, which are after all very expensive, if people refuse to take advantage of the free vaccination. The vaccines have been strictly controlled by the European regulatory authority, are approved in our country, are safe and effective—and I think everyone should get vaccinated if at all possible.'"; *Die Presse*, 5 August 2021)

Finally, Dorothee Von Laer's comments regarding vaccination are quite varied and include comments on cross vaccination (*Der Standard*, 10 August 2021), annual COVID vaccinations in the future (*Die Presse*, 18 February 2021), or the 2-G rule (proof of vaccination or recovery to go to restaurants, the theater, etc.; *Der Standard*, 29 August 2021). Most notably, despite conflicts of interest, she proposes that recovery from a previous COVID infection protects better against reinfection than the vaccines, as example (18) shows:

18.    „Dass man mit einer 1-G-Regel auch Genesene ausschließe, ergebe ‚überhaupt keinen Sinn', meinte zuletzt die Virologin Dorothee von Laer (Medizin-Uni Innsbruck) im Magazin ‚Profil'. Schließlich seien Genesene vor einer—neuerlichen—Ansteckung sogar besser geschützt als Geimpfte." ("'It makes 'no sense at all' to exclude convalescents with a 1-G rule, said the virologist Dorothee von Laer (Innsbruck Medical University) in the magazine 'Profil'. After all, those who have recovered are even bet-

ter protected against a—renewed—infection than those who have been vaccinated."; *Die Presse*, 30 August 2021)

Overall, we can see a mixed picture when it comes to conflicts of interest. While Krammer and Szekeres fully follow the pharma industry's 'safe and effective' narrative, Czypionka and Von Laer, on occasion, voice viewpoints that do not follow big pharma's storyline.

*3.5. Critical Comments*

The only clearly critical expert comments, often introduced through the reporting verb *criticized*, come from Martin Sprenger, who was a member of Austria's Corona Task Force and thus a media-approved expert for only a few months. Sprenger mainly criticizes fear-mongering (*Der Standard*, 13 May 2020), the government's heavy reliance on medical experts to the exclusion of social scientists, psychologists, and educators (*Der Standard*, 15 April 2020; *Der Standard*, 26 April 2020), as well as a lack of focus on how COVID-19 measures may affect vulnerable populations, such as children (*Der Standard*, 15 April 2020). He clearly points out the negative effects that these factors likely have had or will have on the population. In examples (19) and (20), for instance, he specifies the potential negative effects of fear mongering:

19. "Dann ein Paper mit 100.000 Toten aus dem Talon zu ziehen war ein riesiger Fehler und eine unnötige Eskalation', so Sprenger. Hätte man sich anders entschieden, wären ‚die Köpfe heute nicht so voller Angst, die Schäden in der Regelversorgung sicher geringer und die Arbeitslosigkeit wahrscheinlich nicht so hoch'." ("To then pull a paper with 100,000 dead from the talon was a huge mistake and an unnecessary escalation,' Sprenger said. Had the decision been made differently, 'heads wouldn't be so full of fear today, the damage to standard care would certainly be less, and unemployment probably wouldn't be so high'."; *Der Standard*, 13 May 2020)

20. „Dies habe möglicherweise mehr Menschenleben gekostet als gerettet, argumentiert Sprenger, weil viele Kranke aus Angst nicht ins Spital gegangen sind und soziale, psychische, pädagogische und ökonomische Schäden in Kauf genommen worden seien. Was fehle, sei eine konsequente 'Gesundheitsfolgenabschätzung' aller Maßnahmen." ("This may have cost more lives than it saved, Sprenger argues, because many sick people did not go to the hospital out of fear, and social, psychological, educational and economic damage was accepted. What is missing, he says, is a consistent 'health impact assessment' of all measures."; *Der Standard*, 11 May 2020)

In examples (21) and (22), he focuses on children and discusses the negative consequences that school closures can have on children, specifically those from lower socioeconomic backgrounds:

21. „Seine spezielle Sorge gilt derzeit den vom Virus kaum betroffenen Kindern, ‚die hoffentlich nicht durch Gewalt in der Familie oder Zwangsmaßnahmen in den wieder zu eröffnenden Schulen Traumata davontragen'. Auf diese Gefahr wird er, selbst Vater zweier Kinder, wohl noch oft hinweisen." ("His special concern at the moment is for the children, a group that is hardly affected by the virus, 'who hopefully will not suffer trauma from violence in the family or coercive measures in the schools that are to be reopened.' He, himself the father of two children, will probably point out this danger often."; *Der Standard*, 15 April 2020)

22. „Public-Health-Experte Martin Sprenger etwa, einst Mitglied des Krisenstabs, empfiehlt dringend, Schulen und Kindergärten wieder aufzusperren. [. . .] Die Warnungen von Lehrerinnen und Lehrern und Direktorinnen und Direktoren, dass sie einen erklecklichen Anteil ihrer Schülerinnen und Schüler, vor allem aus sozial schwächeren Familien, gar nicht mehr erreichen, die Hilferufe von Eltern, die mit der Heimbeschulung überfordert sind, bleiben von Regierungsseite unkommentiert." ("Public health expert Martin Sprenger, for example, once a member of the crisis team, urgently recommends reopening schools and kindergartens. [. . .] The warnings of teachers and principals that they no longer reach a considerable proportion of their students,

especially those from socially weaker families, and the cries for help from parents who are overwhelmed by home schooling, remain uncommented on by the government.";
*Der Standard*, 18 April 2020)

As the examples show, Sprenger is rather concerned about the most vulnerable people in society. Interestingly, Sprenger often voices moral considerations, and it is thus surprising that Christiane Druml, the ethicist among the experts, did not voice similar concerns. Instead of considering individuals, her focus is instead on (the vaccinated) society as a whole, with little consideration for individuals' particular needs and situations, especially if these individuals are unvaccinated. She is one of the most vocal proponents of a country-wide vaccine mandate (*Der Standard*, 19 December 2020), for unvaccinated people paying extra for hospital care (*Der Standard*, 16 September 2021), as well as an outspoken opponent of free speech during the COVID crisis. Examples (23) and (24) illustrate her general stance.

23. „Sollen in diesem Fall ungeimpfte Covid-Kranke anderen Patienten wirklich gleichgestellt werden? Die medizinische Ethik verlangt dies, betont Druml. Noch mehr: Laut einem Dokument der Bioethikkommission sollte im Notfall ‚die kurzfristige Überlebenswahrscheinlichkeit der maßgebliche Orientierungspunkt' für die Entscheidung sein, wer zuerst behandelt wird. Aber das würde bedeuten, dass ein junger Covid-Kranker, der die Impfung verweigert hat, einer älteren Krebspatientin vielleicht vorgezogen werden muss—ein schlimmes ethisches Dilemma. Drumls bedrückendes Fazit: ‚Wir wollen alle wieder ein normales Leben führen, aber ich weiß kein Rezept, wie das möglich wird, wenn sich diese Menschen nicht impfen lassen.'" ("In this case, should unvaccinated Covid patients really be treated the same as other patients? Medical ethics demands this, Druml emphasizes. More to the point, according to a document from the Bioethics Committee, in an emergency, 'the short-term probability of survival should be the relevant point of reference' for deciding who gets treated first. But that would mean a young Covid patient who refused vaccination might have to be given preference over an older cancer patient—a dire ethical dilemma. Drumm's depressing conclusion: 'We all want to get back to a normal life, but I don't have a recipe for how that's going to be possible if these people don't get vaccinated.'"; *Der Standard*, 11 September 2021)
24. „Da fehlt einem das Verständnis, dass eine völlig desinformierte, völlig die Tatsachen ignorierende Meinung vorhanden ist und die in der Öffentlichkeit laut vorgetragen wird. Das und unverständliche Protestaktionen vor Spitälern sind eine Belastung für die Gesellschaft. Und das muss bereinigt werden." ("I have no sympathy for the fact that a completely disinformed opinion that completely ignores the facts exists and is loudly expressed in public. That and incomprehensible protests in front of hospitals are a burden on society. And that needs to be rectified."; *Der Standard*, 20 December 2021)

An exception to this is example (25), where she supports keeping schools open as long as possible, thus showing concern for children's rights to an education.

25. „STANDARD: Sollten dann die Schulen um jeden Preis offen bleiben?" „Druml: Nicht um jeden Preis, aber so lange wie möglich und mit einem Betrieb so normal wie möglich." ("STANDARD: Should schools then remain open at all costs?" "Druml: Not at all costs, but for as long as possible and with operations as normal as possible."; *Der Standard*, 20 December 2021)

Despite some arguably rather unethical statements, Druml was treated as an official expert until the end of the pandemic; Sprenger, on the other hand, was soon discarded and criticized, and later on even slandered by certain media outlets, for instance, for giving an interview to an alternative right-wing news portal (Tschiderer 2022).

## 4. Discussion

The analysis above reveals several interesting aspects. First and foremost, the main experts that were consulted during the COVID-19 event consistently and almost exclusively

promoted the government's narrative, which confirmed the findings by Hollar (2022), Herman and Chomsky (2002), and Debord (1988), and also supported Bennett's (1990) claim that the 'marketplace of ideas' dissipates in times when it would be needed most. This is perhaps unsurprising, as they were mostly members of government-linked task forces starting in April 2020. The only expert who dissented from the official position and questioned the dominant narrative, Martin Sprenger, soon left the task force, which is perhaps the best evidence to suggest that dissent was not seen as acceptable and that by leaving the task force and dissenting from the mainstream, Sprenger inadvertently lost his expert status. This is in line with a study on scientific experts in the media by Neresini et al. (2023), which found that "being a *media star* is not an irreversible achievement." By drawing on opinions from individuals that echoed and complemented each other's statements, an illusion of consent was created, which made any diverging claim or narrative seem fringe and outrageous.

Not only was the selection of experts one-dimensional, but our analysis of reporting verbs has also shown that their claims were taken at face value (research question 2), which we found to be in line with the findings and claims by Sigal (1986) and Carlson (2012). In our corpus, we did not detect any distancing devices or indicators of uncertainty—with the exception of prognoses. This is clearly problematic as it "contradicts the logic of science, which is actually characterized by uncertainty and tentativeness" (Nguyen and Catalan-Matamoros 2020; cited by Nölleke et al. 2023). Whatever the experts postulated, it was imbued by journalists with authority and credibility by virtue of the expert label. In part, this may be due to the "mutually beneficial relationship" (Carlson 2012) and journalists' interest in maintaining good working relationships with their sources (cf. Sigal 1986). It may also be owing to the sources' role as "provid[ing] authority to journalistic products" (Wien 2014, p. 440), due to having been selected in order to "confirm and legitimize a news frame" (Albaek 2011, p. 344). Indeed, a study by Wien (2001) has shown that only 1% of news items analyzed took a critical stance towards the expert, while some 65% had an uncritical attitude, simply because experts were chosen to confirm the news frame in the first place. By quoting only conformist experts and by treating their information as virtually sacrosanct ('the science'), the media allowed them to "defin[e] [. . .] consensus at a high level" (Kissinger 1966, p. 514) to the exclusion of any unorthodox opinions, which as a consequence were—if not outright ignored—attached with deprecatory labels such as those mentioned in our introduction.

Yet another dimension of the expert problem was revealed through our analysis of expertise (research question 3). In contrast to Ioannidis et al.'s (2021) results, the most highly visible experts in the Austrian media that we analyzed had mostly published on COVID-19 and, in line with Leidecker-Sandmann et al.'s (2022) results, thus had a high level of expertise. However, the corpus also showed that, while some of the experts more or less stayed within their realm of expertise, there were others who—with the authority vested in them by the media—commented on issues for which they did not have the necessary credentials, thus supporting Boyce's conclusion that there is a need for journalists to assess their sources' expertise more and enhance source transparency (2007, p. 903). Thus, simulation researchers gave their opinions on the complex connection between vaccination rates and infection rates; complex system researchers emphasized the importance of lockdowns and masking; and ethicists talked about the dangers posed by 'the unvaccinated'. In the absence of any critical comments by the journalists, readers are very likely to have taken these claims at face value and to have seen them as credible; after all, they are coming from someone the media call an 'expert'. An expert, perhaps, but only such within the narrow constraints of their respective fields.

Finally, our research on potential conflicts of interest has yielded some notable results as well (research question 4). In line with previous research (Murayama et al. 2021), most experts had conflicts of interest. The four experts with overt connections to the pharma industry commented extensively on the vaccines, overwhelmingly promoting the message that the vaccines are "safe and effective". This is interesting since other studies, such as

Wien (2014, p. 434), have found a loatheness to use "industrial sources" due to questions about their authority and credibility because of their professional links. We also found some division of labor, where different experts focused on different aspects of the "safe and effective" message. For example, while Krammer focused on telling the public that the vaccines are safe, Czypionka highlighted their effectiveness. Another preoccupation of the experts with conflicts of interest with respect to the pharma industry was how to increase the number of people vaccinated. Here, Czypionka and Szekeres favored different approaches. While Czypionka wanted to reach the unvaccinated and convince them to get vaccinated, Szekeres strongly promoted vaccine mandates and measures that actively discriminate against the unvaccinated. Thus, for the most part, experts with conflicts of interest promoted messages that were in line with the financial interests of the pharma industry.

**5. Conclusions**

The media clearly play a crucial role in selecting who will be seen as an 'expert' by the public. This choice, our analysis indicates, is arguably not exclusively due to any deep knowledge on the subject nor to the individual being more prolific in their field than others, and thus paints a slightly different picture than Leidecker-Sandmann et al.'s 2022 study on COVID experts in the German media, which is more in alignment with the questions raised by Boyce (2007). To the extent that we can generalize, based on the corpus of the two Austrian broadsheets we analyzed, it is clear that the expert label has much more to do with whether or not a person is likely to embrace the predominant narratives and talking points propagated by people in power. This is also in line with similar findings by Albaek (2011) and Wien (2014), showing that journalists usually contact researchers who will confirm and legitimize their news frame. The fact that the media outlets we analyzed chose—largely identical—COVID experts with direct links to government task forces is a further indication of this as well as of a questionable bandwagon effect at the expense of diversity of expert voices. That many of the selected experts had—sometimes significant—conflicts of interests, promoting courses of action that would directly benefit their funders or their own research, careers, and companies, was another red flag. It was a red flag in particular because the media failed to highlight these conflicts of interest and thus allow the readers to contextualize what these experts were advocating for and make up their own minds. The media also permitted and invited anyone from their pool of experts to comment on a whole range of issues, at times far removed from their expertise. By providing any expert that happened to advance official narratives with a stage to comment on anything COVID, the media have inarguably both devalued and demasked the expert label[1]. If anybody can be an expert if only they have the 'right' worldview, if any expert can comment on any issue beyond their expertise with the confidence of someone in the know, then the title of 'expert' becomes an empty shell, which is awarded to people based on their conformity with and obedience to the agendas of those in power. The age of the expert, which Henry Kissinger so emphatically declared decades ago, is clearly upon us, but in this age of the expert, it might be the voice of the puppet master that we are hearing.

**Author Contributions:** Conceptualization, J.S. and A.F.; methodology, J.S. and A.F.; formal analysis, J.S. and A.F.; investigation, J.S.; data curation, J.S. and A.F.; writing—original draft preparation, J.S. and A.F.; writing—review and editing, J.S. and A.F.; project administration, J.S. All authors have read and agreed to the published version of the manuscript.

**Funding:** This research received no external funding.

**Institutional Review Board Statement:** Not applicable.

**Informed Consent Statement:** Not applicable.

**Data Availability Statement:** The data presented in this study are available on request from the corresponding author. The data are not publicly available due to paywall restrictions.

**Acknowledgments:** We would like to thank Katharina Haslacher for her help with building and cleaning the corpus.

**Conflicts of Interest:** The authors declare no conflicts of interest.

## Note

[1] This also seems to be confirmed by studies listed by Nölleke et al. (2023, p. 548) that attest to a dramatic mistrust in science, as well as a study by Larsson et al. (2019, p. 9), which found "experts willing to comment on stories in spite of conflict of interest", leading to an erosion of public trust in science.

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
