# Peer review of "The Age of the Expert—COVID-19, Expertise, and Conflicts of Interest in Austrian Media Reporting"

_journalmedia, doi:10.3390/journalmedia5010012_

Round 1

Reviewer 1 Report

Comments and Suggestions for Authors

This is a very well conducted piece of research and should be published. The conceptual/theoretical underpinnings are sound and a good familiarity with the political communication literature concerning sources and news production is on show here. Broadly, the theoretical context for this paper is the indexing paradigm (Bennett, 1990) but I am glad to see the author(s) engage with a wide and rich literature regarding journalistic reliance upon, and deference to, authority.

The research approach is particularly well done. It combines a systematic and reproduceable component which includes an automated approach to assessing the stance adopted by journalist towards the top 10 cited COVID-19 sources; and then offers a more in depth qualitative reading of some of the comments made by the sources in relation to their areas of expertise. The result is a well-evidenced and therefore persuasive account of media deference to authority.

Another strong aspect of this study is the application to the health care issue area and the recent COVID-19 event. The study provides a test of the applicability of the indexing paradigm to this issue area and a recent case.

Finally, in the context of the current emerging controversies regarding the COVID-19 response, this study serves as a reminder of the problems associated with journalistic deference to official sources and the fact that this problem has not gone away.   

Minor Point: Avoid pejorative descriptors language such as ‘pushing the government line’. 

Author Response

This is a very well conducted piece of research and should be published. The conceptual/theoretical underpinnings are sound and a good familiarity with the political communication literature concerning sources and news production is on show here. Broadly, the theoretical context for this paper is the indexing paradigm (Bennett, 1990) but I am glad to see the author(s) engage with a wide and rich literature regarding journalistic reliance upon, and deference to, authority.

Thank you! We have now also added explicit references to the indexing paradigm to make the link here more apparent.

The research approach is particularly well done. It combines a systematic and reproduceable component which includes an automated approach to assessing the stance adopted by journalist towards the top 10 cited COVID-19 sources; and then offers a more in depth qualitative reading of some of the comments made by the sources in relation to their areas of expertise. The result is a well-evidenced and therefore persuasive account of media deference to authority.

Another strong aspect of this study is the application to the health care issue area and the recent COVID-19 event. The study provides a test of the applicability of the indexing paradigm to this issue area and a recent case.

Finally, in the context of the current emerging controversies regarding the COVID-19 response, this study serves as a reminder of the problems associated with journalistic deference to official sources and the fact that this problem has not gone away.   

Thanks so much for the positive feedback!

Minor Point: Avoid pejorative descriptors language such as ‘pushing the government line’. 

Thank you for this advise. We have removed traces of what could be interpreted as pejorative language.

Reviewer 2 Report

Comments and Suggestions for Authors

The manuscript makes an important contribution to the existing knowledge on the topic and, as a result, is relevant for the field. 

The text is well-structured and uses extended and appropriate references.

In the materials and method section, the authors should specify how the ten most frequently mentioned experts are distributed among the two newspapers selected (are the majority more mentioned by Der Standard or Die Presse, or both?). Moreover, it is necessary to justify the selection of these two newspapers: why two and these two instead of others? (if they are the most read, for example). The authors should notice that many readers are probably not familiar with Austrian media.

On the other hand, when the authors talk about their investigation on the experts’ professional background they don’t reveal the employed method or concrete ways they have carried out the research. They must be as much specific as they can within this section.  

It would be necessary to repeat/insist on the main aim/objective of the article (it is introduced in the previous paragraph, but not reflected here). Including several hypotheses might be helpful for that purpose.

In the discussion and conclusions sections, the authors forget to reflect on the way that journalists normally select the experts they will consult for their articles (journalists previously filter the experts they consult due to their expertise, the relation of trust built between both, etc.). All this might be key to understanding why “say” is the most used reporting verb. In addition, the fact that the pandemic has been the most significant challenge humanity has confronted in the last century in terms of health but also information should be highlighted. Journalists (as well as scientists) faced uncertainty and a clear lack of information, which probably conditioned the way and methods they used to select the sources and, most importantly, their attitude and perceptions when writing Covid-19-related articles. Quoting experts was probably the only way –apart from consulting academic papers– journalists had to approach this reality in the midst of uncertainty.

Author Response

The manuscript makes an important contribution to the existing knowledge on the topic and, as a result, is relevant for the field. 

The text is well-structured and uses extended and appropriate references.

Thanks so much for the positive feedback!

In the materials and method section, the authors should specify how the ten most frequently mentioned experts are distributed among the two newspapers selected (are the majority more mentioned by Der Standard or Die Presse, or both?). Moreover, it is necessary to justify the selection of these two newspapers: why two and these two instead of others? (if they are the most read, for example). The authors should notice that many readers are probably not familiar with Austrian media.

Thanks so much for this comment. We have added information to the paper to justify our selection of these two newspapers. Specifically, we selected these newspapers because they are considered “trustworthy” and they cover a broad political spectrum. We have also added a breakdown for the number of times the experts were quoted in each newspaper in table 1. We hope that this will now provide a more comprehensive overview and insight.

On the other hand, when the authors talk about their investigation on the experts’ professional background they don’t reveal the employed method or concrete ways they have carried out the research. They must be as much specific as they can within this section.  

Thanks so much for this comment. We have added information as to how we did our investigation of the experts’ professional backgrounds.

It would be necessary to repeat/insist on the main aim/objective of the article (it is introduced in the previous paragraph, but not reflected here). Including several hypotheses might be helpful for that purpose.

Thank you for this comment. We have made the main aim of the article more explicit and we have also added research questions (rather than hypotheses) to the text. We pick up the research questions again in the results section and now also refer to them in the discussion section. We hope that this suffices to emphasize the main aim of the article.

In the discussion and conclusions sections, the authors forget to reflect on the way that journalists normally select the experts they will consult for their articles (journalists previously filter the experts they consult due to their expertise, the relation of trust built between both, etc.). All this might be key to understanding why “say” is the most used reporting verb.

Thank you for raising this issue. We have added a paragraph on how journalists choose their sources in the introduction and commented on this in our conclusion as well.

In addition, the fact that the pandemic has been the most significant challenge humanity has confronted in the last century in terms of health but also information should be highlighted. Journalists (as well as scientists) faced uncertainty and a clear lack of information, which probably conditioned the way and methods they used to select the sources and, most importantly, their attitude and perceptions when writing Covid-19-related articles. Quoting experts was probably the only way –apart from consulting academic papers– journalists had to approach this reality in the midst of uncertainty.

We have tried to highlight how the choice of experts was still biased, seeing that media outlets used largely the same experts and that they all agreed on basically the major issues. We discussed Martin Sprenger as an example of an expert that gets his credentials revoked due to his antagonizing opinions. The fact is, journalists had a choice of various experts, but chose to go with those that were ‘safe’.

Reviewer 3 Report

Comments and Suggestions for Authors

Dear Authors,

thank you for the opportunity to read this thought-provoking article on the use of expert sources in Austrian media during COVID-19. I absolutely agree that it is a worthwhile endeavor to examine the journalistic construction of expertise, and I appreciate the authors' critical attitude towards this phenomenon. I don't endorse every single aspect of their (at times) normative argument (for example, I’m not sure that media indeed strive to delegitimize “Coronaleugner” by “suggesting they are ready to fall for whatever the newest conspirancy fad is”, p. 3, ll. 118; I would also not necessarily agree that it is “more likely than not that their financial involvement with said companies affects the content of their media comments”, p. 4, ll. 178) - but then again: it's not for me to decide on the legitimacy of such claims; that's a question that should be addressed empirically.

Yet this is precisely where my greatest concerns arise. A potential strength of this paper is that it could very well contribute to and advance existing research on the relationship between journalists and (scientific) experts. Unfortunately, however, you do not refer to relevant findings by, for example, Albaek (2011), Albaek et al. (2014), Huber (2013), Leidecker-Sandmann et al. (2012), Nölleke et al. (2023), Peters (2013), Prommer & Stüwe (2020), Shine (2021), Wien (2014) etc. While you cite some relevant authors (like Soley) who have studied the use of expert sources in broader journalism, you fail to show how your study relates to recent research on science communication. Thus, it remains completely unclear how your work contributes to existing research. This does not mean at all that I want to ask you to comply with previous knowledge - but it would have been necessary to benefit from it and to highlight your own contribution against this background.

Furthermore, I have grave concerns about the presentation of the findings. In qualitative research it is essential to give examples - this is true. However, large parts of the results section consist entirely of quotations linked by only one or two sentences. It would be necessary to analyze this data more expertly, for example by paraphrasing or/and identifying themes.

Since this work does not meet the crucial standards for the preparation of literature reviews and the analysis of qualitative data, I cannot recommend it for publication. In addition to these main concerns, I have several others related to, for example, the operationalization of the research interest, the (lack of) explanation of the case selection, etc. However, since this work (on such a promising and important topic) does not (in my opinion) meet basic research standards, I will not go into these details for now.

I wish you all the best for your (interesting) work on this (important) and other topics.

Author Response

Dear Authors,

thank you for the opportunity to read this thought-provoking article on the use of expert sources in Austrian media during COVID-19. I absolutely agree that it is a worthwhile endeavor to examine the journalistic construction of expertise, and I appreciate the authors' critical attitude towards this phenomenon. I don't endorse every single aspect of their (at times) normative argument (for example, I’m not sure that media indeed strive to delegitimize “Coronaleugner” by “suggesting they are ready to fall for whatever the newest conspirancy fad is”, p. 3, ll. 118; I would also not necessarily agree that it is “more likely than not that their financial involvement with said companies affects the content of their media comments”, p. 4, ll. 178) - but then again: it's not for me to decide on the legitimacy of such claims; that's a question that should be addressed empirically.

Thanks for this comment. We have slightly rephrased the sentence about delegitimizing Coronaleugner so as not to suggest that the purpose of these labels is obvious. We have also rephrased the first quote as “by suggesting they are particularly susceptible to believe in conspiracy theories of any kind” and we have repeated relevant citations to support the second quote.

Yet this is precisely where my greatest concerns arise. A potential strength of this paper is that it could very well contribute to and advance existing research on the relationship between journalists and (scientific) experts. Unfortunately, however, you do not refer to relevant findings by, for example, Albaek (2011), Albaek et al. (2014), Huber (2013), Leidecker-Sandmann et al. (2012), Nölleke et al. (2023), Peters (2013), Prommer & Stüwe (2020), Shine (2021), Wien (2014) etc. While you cite some relevant authors (like Soley) who have studied the use of expert sources in broader journalism, you fail to show how your study relates to recent research on science communication. Thus, it remains completely unclear how your work contributes to existing research. This does not mean at all that I want to ask you to comply with previous knowledge - but it would have been necessary to benefit from it and to highlight your own contribution against this background.

Thanks so much for this comment and for the literature suggestions. We have incorporated all the literature that is directly relevant to our paper in the revised document. Specifically, we now refer to Albaek (2011), Albaek et al. (2014), Leidecker-Sandmann et al. (2022), Nölleke et al. (2023), Prommer & Stüwe (2020), Shine (2021), and Wien (2013).

Furthermore, I have grave concerns about the presentation of the findings. In qualitative research it is essential to give examples - this is true. However, large parts of the results section consist entirely of quotations linked by only one or two sentences. It would be necessary to analyze this data more expertly, for example by paraphrasing or/and identifying themes.

Thanks so much for this comment. We have now identified themes and paraphrased some of the quotes, and we hope that this has improved the results section.

Since this work does not meet the crucial standards for the preparation of literature reviews and the analysis of qualitative data, I cannot recommend it for publication. In addition to these main concerns, I have several others related to, for example, the operationalization of the research interest, the (lack of) explanation of the case selection, etc. However, since this work (on such a promising and important topic) does not (in my opinion) meet basic research standards, I will not go into these details for now.

We have made major changes to the literature review and the qualitative data analysis, based on the suggestions above, and we hope that the paper is now acceptable for publication. We hope that the addition of research questions and additional detail in the methods section has clarified how we have operationalized our research interest. We have also added information about why we selected these particular two newspapers, and we hope that this provides a sufficient explanation of our case selection.

I wish you all the best for your (interesting) work on this (important) and other topics.

Round 2

Reviewer 3 Report

Comments and Suggestions for Authors

Thank you for the opportunity to read the revised version of this paper. I really appreciate how the authors responded to my comments. It is good that they have included some literature on the use of expert sources in journalism; and I also believe that the findings section has been improved by removing some of the extensive citations. However, my main concern remains: This is basically an article about the transparency of expert sources, right? And it suggests that the media has a bias that is served by experts (repeating the prevailing narrative). However, neither this media bias nor the issue of source transparency is introduced by reference to academic literature. Furthermore, the literature on the use of expert sources is mainly used to show which topics have already been researched, but less to build one's own analysis on. In my opinion, this is not taken as seriously as it should have been in order to highlight the precise contribution of this paper (which is of course still interesting). So what area of research does it inform, what theoretical contribution does it make? This particularly relates to RQ3 and RQ4. The authors suggest that sources simply comment on issues that are beyond their specific expertise. However, we know from previous research that they are particularly asked to do so because journalists lack time, resources, and competencies to always identify the expert with the most superior knowledge. This all points in the direction that experts are the result of a journalistic construction process – and that the intended bias of an article is only one aspect among others that influences source selection. However, the authors suggest that it is just the intention to strengthen the dominant narrative. Doing this they fail to refer to (potential) alternative mechanisms of journalists’ employment of expert sources; and they – as mentioned above – fail to refer to literature that supports their assertion of a media bias.

I really appreciate how the authors discuss their findings in the last sections of this paper and how they relate them to previous research (the discussion is far better now) - however, they fail to derive their own research interest from previous research.

While the article does not convincingly demonstrate how it advances related academic literature, it builds its own narrative primarily on assertions unsupported by empirical evidence. In stating that “not surprisingly, the financial or professional ties of experts co-determines the content and line of reasoning of their comments” (p. 2, l. 65) the authors do not refer to any literature at all; in stating “a recent trend has seen journalists and opinion writers become multipliers and promotors of pharma products, without even an inch of critical scrutiny of the product they report on” (p. 4, ll. 194) the authors merely refer to some selected media articles; and when they write “What this means for medical experts quoted in the media is obvious: if they have ties to or receive funding from the pharmaceutical companies whose products they are called to comment on, it is more likely than not that their financial involvement with said companies affects the content of their media comments” (p. 5, ll. 209) they are also not referring to actual research on media appearances. I simply get the impression that their strong claims about the bias of expert sources cannot be supported by empirical evidence.

What is more, when the authors refer to balanced reporting, they fail to acknowledge the research on objectivity and false balance; and I disagree with their use of "pseudo-experts" - at the very least, the term should have been explained and defined, as it is very commonly used in research on the media's use of expert sources. There are also other critical aspects: For example, the authors should have elaborated a bit more on the relationship between media experts and politics. They seem to suggest that most of the experts were already members of official working groups; however, as far as I know, some of them only became members through their media appearances. This relationship between media and politics (which is well worth exploring) remains completely under-examined in this article.

To summarize, while I see some improvements, I am still not convinced by the first part of the article, as the relevant literature is not directly used and important claims are not supported by empirical evidence (which is disguised by heavy reference to news articles or popular books).

The same applies to the presentation of the results. There are huge improvements, but the authors still rely on very long direct quotes, one below the other. It is good that they paraphrase the most important results. Still, this way of presenting results is not what I expect from qualitative research. I would recommend avoiding too extensive quotes and especially not listing them one below the other.

Apart from these main concerns, there are a few others (e.g. the extensive use of "cited in"). Most importantly, I continue to see an urgent need to base strong claims about expert bias and media bias on more solid literature.

Round 3

Reviewer 3 Report

Comments and Suggestions for Authors

First of all, I would like to take this opportunity to thank the authors (again) for their very thorough response and revisions. I believe that the paper is much better now - I am particularly convinced by the literature review. Perhaps two more comments: I like that the authors have now emphasized objectivity even more. However, I don't agree that the norm of objectivity forces sources to be subjective (which is now the very first argument of the paper). To avoid subjectivity, journalists don't necessarily need sources. But that's just a minor point. The same goes for my impression that the paragraph about the media's reference to covidiots. is still not supported by empirical evidence and promotes a rather media-skeptical narrative.
I must admit that I still have my doubts about the presentation of the results of the qualitative analysis (extensive quotes), but the authors explain this well and I would not ask them to change this.
Nevertheless, there is one aspect that I would suggest to revise. This article is about experts and consequently people are mentioned who are presented as experts. However, if I am not mistaken, the methods section does not explain what criteria the selected expert sources had to fulfill in order to be included in the analysis. Were they actually referred to as experts by the media? How was this authority constructed by the media?
I also get the impression that the authors are not only suggesting that the experts are not capable of providing expert opinions (due to a lack of expertise), but that their statements are flawed due to their industry affiliation. I understand this assertion - and I think the results are ambivalent and not idiosyncratic (as I had previously criticized). Still, couldn't it be that experts with ties to the industry made correct statements? I have the feeling that the authors are suggesting that they were wrong; e.g. by using the term "overestimated" (p. 18, line 794). But: what is the standard for calling this an overestimation? So, perhaps the connection to industry is evidence of expert knowledge and this expert knowledge is the basis for being presented in the media as an expert source. It's not at all that I wouldn't require the media to be much more transparent about industry connections. It's more that I'm skeptical as to whether such connections are automatically problematic when an expert represents the industry's position. It could still be right, right? Because the industry is not necessarily evil.

I really think this paper has improved, even if I still have some concerns about some (implicit) underlying assumptions. It deals with an interesting case and raises relevant questions. The results are interesting. Before publishing the paper, I would suggest that the authors clarify their expert selection criteria and perhaps think about defusing some of the implicit assumptions that links to industry inevitably lead to biased (and incorrect) statements.

Author Response

First of all, I would like to take this opportunity to thank the authors (again) for their very thorough response and revisions. I believe that the paper is much better now - I am particularly convinced by the literature review.

Thanks so much for the positive feedback.

Perhaps two more comments: I like that the authors have now emphasized objectivity even more. However, I don't agree that the norm of objectivity forces sources to be subjective (which is now the very first argument of the paper). To avoid subjectivity, journalists don't necessarily need sources. But that's just a minor point.

We have changed the wording a little here, so that we are not suggesting that journalists necessarily need sources.

The same goes for my impression that the paragraph about the media's reference to covidiots. is still not supported by empirical evidence and promotes a rather media-skeptical narrative.

We have now shortened the information on these kinds of references and also weakened the language, so that it is no longer as strong of a claim. We hope that this is ok now.  

I must admit that I still have my doubts about the presentation of the results of the qualitative analysis (extensive quotes), but the authors explain this well and I would not ask them to change this.

Nevertheless, there is one aspect that I would suggest to revise. This article is about experts and consequently people are mentioned who are presented as experts. However, if I am not mistaken, the methods section does not explain what criteria the selected expert sources had to fulfill in order to be included in the analysis. Were they actually referred to as experts by the media? How was this authority constructed by the media?

Thanks for pointing this out. We have now refined the criteria we used to select the experts in our corpus further. In essence, we clarify that these had to be explicitly called ‘experts’ by the respective media and they had to occur regularly during the time in question and inform the debate sufficiently to be likely known to the audience.

I also get the impression that the authors are not only suggesting that the experts are not capable of providing expert opinions (due to a lack of expertise), but that their statements are flawed due to their industry affiliation. I understand this assertion - and I think the results are ambivalent and not idiosyncratic (as I had previously criticized). Still, couldn't it be that experts with ties to the industry made correct statements? I have the feeling that the authors are suggesting that they were wrong; e.g. by using the term "overestimated" (p. 18, line 794). But: what is the standard for calling this an overestimation? So, perhaps the connection to industry is evidence of expert knowledge and this expert knowledge is the basis for being presented in the media as an expert source. It's not at all that I wouldn't require the media to be much more transparent about industry connections. It's more that I'm skeptical as to whether such connections are automatically problematic when an expert represents the industry's position. It could still be right, right? Because the industry is not necessarily evil.

Thank you for this comment. Our use of “overestimated” is in reference to one of the experts assuming that the protection from the Covid-19 vaccines will last for several years. Since it turned out that booster shots were needed and since many countries, including Austria, now recommend annual vaccinations for Covid-19, this seems to be an overestimation. To address that experts with industry ties are not necessarily wrong, we have added the sentence “While claims of experts with ties to industry or official organizations are not necessarily wrong, concerns about such experts’ objectivity are widespread and the evidence that does exist suggests that these concerns are warranted (Bailey et al. 2011; De Dobbelaer et al. 2017; Goldacre 2014; Lexchin 2012; Moynihan et al. 2020; Roussel and Raoult 2020; Sismondo 2021)” to the introduction section. We hope that this clarifies that we are not suggesting that the experts are necessarily wrong. The additional references serve to illustrate that there are widespread concerns about experts with ties to the medical industry.

I really think this paper has improved, even if I still have some concerns about some (implicit) underlying assumptions. It deals with an interesting case and raises relevant questions. The results are interesting. Before publishing the paper, I would suggest that the authors clarify their expert selection criteria and perhaps think about defusing some of the implicit assumptions that links to industry inevitably lead to biased (and incorrect) statements.

Thanks so much. We have addressed these two concerns as well as the other comments above.